# Transcriptome Landscape Variation in the Genus *Thymus*

**DOI:** 10.3390/genes10080620

**Published:** 2019-08-16

**Authors:** Aboozar Soorni, Tahereh Borna, Ali Alemardan, Manohar Chakrabarti, Arthur G. Hunt, Aureliano Bombarely

**Affiliations:** 1Department of Biotechnology, College of Agriculture, Isfahan University of Technology, Isfahan 84156-83111, Iran; 2Department of Horticultural Science, College of Agriculture and Natural Resources, University of Tehran, Karaj 31587-11167, Iran; 3Department of Plant and Soil Sciences, University of Kentucky, Lexington, KY 40546-0312, USA; 4School of Plant and Environmental Sciences, Virginia Tech, Blacksburg, VA 24060, USA; 5Department of Bioscience, University of Milano, 20133 Milan, Italy

**Keywords:** *Thymus*, terpenoid biosynthesis, transcriptome, phylotranscriptome

## Abstract

Among the *Lamiaceae* family, the genus *Thymus* is an economically important genera due to its medicinal and aromatic properties. Most *Thymus* molecular research has focused on the determining the phylogenetic relationships between different species, but no published work has focused on the evolution of the transcriptome across the genus to elucidate genes involved in terpenoid biosynthesis. Hence, in this study, the transcriptomes of five different *Thymus* species were generated and analyzed to mine putative genes involved in thymol and carvacrol biosynthesis. High-throughput sequencing produced ~43 million high-quality reads per sample, which were assembled de novo using several tools, then further subjected to a quality evaluation. The best assembly for each species was used as queries to search within the UniProt, KEGG (Kyoto Encyclopedia of Genes and Genomes), COG (Clusters of Orthologous Groups) and TF (Transcription Factors) databases. Mining the transcriptomes resulted in the identification of 592 single-copy orthogroups used for phylogenetic analysis. The data showed strongly support a close genetic relationship between *Thymus vulgaris* and *Thymus daenensis.* Additionally, this study dates the speciation events between 1.5–2.1 and 9–10.2 MYA according to different methodologies. Our study provides a global overview of genes related to the terpenoid pathway in *Thymus*, and can help establish an understanding of the relationship that exists among *Thymus* species.

## 1. Introduction

*Thymus* is one of the most consequential genera belonging to the *Lamiaceae* family, and contains more than 336 species of aromatic perennial herbaceous plants with valuable medicinal properties [1,2]. Most species originated in the Mediterranean region but have since have been distributed worldwide [3]. Several members of the genus are also cultivated as culinary herbs or for ornamentation [4].

Many studies have investigated the chemical composition and biological properties of essential oils in different species of *Thymus* [5,6]. The six major constituents showing variations among *Thymus* species include thymol, carvacrol, γ-terpinene, geraniol, linalool, and p-cymene, [5,6,7]. In regard to previous research, the most important factors that strongly affect the quantities and chemical compositions of essential oils are species, environmental parameters, growth regions, cultivation practices, and plant growth stages [5,6,7]. Thymol has been identified as the major constituent in *Thymus vulgaris* [7,8] *Thymus daenensis* [9,10], and *Thymus lancifolius* [10], while carvacrol is known to be major component in *Thymus pubescens* [11,12] and *Thymus persicus* [13].

Since the introduction of next generation sequencing technology (NGS), there has been a major transformation due to the possibility to extracting abundant genetic information from biological systems through the genome and transcriptome sequence data of any species. NGS has made it possible, in a single sequencing process, to generate millions of genomic DNA or transcriptomic fragments in record time and at ever-lowering costs. NGS has catalyzed a number of important breakthroughs in fields such as agriculture or evolutionary science [14,15]. This tool provides a powerful method of developing molecular markers, studying evolutionary processes, and discovering natural biosynthetic pathways in plants, especially in species with unknown genomes. De novo transcriptome assembly is a critically important approach to exploring gene sequences in organisms without genome sequences [16], and recent progress in RNA-sequencing techniques has provided a fast, reliable, and affordable framework for transcriptomic studies [17].

Since the molecular biology underlying the terpenoid pathway in the genus *Thymus* has not been sufficiently investigated yet, RNA-Seq technology was applied to five different species to explore (a) the molecular basis and potential genes involved in terpenoid biosynthesis, with a focus on thymol and carvacrol biosynthesis and transcription factor (TF) families; and (b) comparative phylogenetic analysis based on single-copy orthogroups, to elucidate the origin of these species. This study will supply basic genetic resources for understanding the biology of the genus *Thymus* and reinforce further research endeavors.

## 2. Material and Methods

### 2.1. Ethics Statement 

All Thymus seeds used in this study were collected from natural fields. No specific permits were required according the Iranian laws, and no endangered, vulnerable, threatened, or protected species were involved.

### 2.2. Plant Material

Seeds of *Thymus* accessions were collected from different locations in Iran during August 2015. The information of the studied samples is shown in Table 1. Sowing seeds was carried out in plastic pots with a perlite–coco peat mixture (1:1 *v*/*v*), then grown in a greenhouse under controlled temperature conditions for 45 days at the greenhouse facilities of Virginia Tech, beginning in April 2016.

### 2.3. Genome Size Measures

Fifty mg of fresh leaf tissue was chopped in 1 mL of DeLaat’s buffer [18] for 20–60 s with the *Capsicum annum* lab strain as an internal standard. Suspensions of nuclei were filtered through 20–50 µm Celltrics filters and centrifuged for 5 min at 200 RPM, then reduced to 200 µl by carefully removing the upper solution. Next, 400 µl of staining solution (propidium iodide (PI)) was added to each tube, mixed well by gently shaking the tubes by hand, then incubated at room temperature in darkness for 20–30 min. Flow cytometry measurement was carried out with a Partec PAS flow cytometer (Partec, http://www.partec.de/, Münster, Germany), then nuclear DNA content was calculated using the following formula [19]:Sample 2C DNA (pg) content = ((Sample G1 peak mean)/(Standard G1 peak mean)) × Standard 2C DNA content (pg DNA).

### 2.4. RNA Extraction, Library Preparation and Sequencing

Based on the genome size pattern, five different species that could cover the whole genome size distribution were chosen for the transcriptome sequencing. Total RNA was extracted from young green leaves for each accession obtained from three different plants using the RNEasy Plant Mini kit (Qiagen, Hilden, Germany), following the manufacturer’s instructions for the library preparation and sequencing. RNA quality and quantity were verified with a Nanodrop spectrophotometer and Agilent 2100 Bioanalyzer.

A fraction of 2 μg of total RNA was used as the starting material for library synthesis using the SMART approach [20]. First, the poly-A mRNA in the total RNA was pulled down using poly-T oligo-attached magnetic beads. After purification, the mRNA was fragmented at 95 °C for 2 min along with RT primer and first strand buffer. The fragmented RNA was used for the synthesis of the first-strand cDNA by adding DTT, dNTPs, RNase inhibitor, and SMARTScribe. Reactions were incubated for 2 h at 42 °C. This was followed by second-strand cDNA synthesis using a template-switching oligonucleotide and an additional 1 μL of SMARTScribe. Subsequently, the cDNAs were purified using two rounds of AMPure bead purification. The samples were amplified using PCR to create the final library. The libraries were confirmed by electrophoresis on a 1% TAE agarose gel and the Agilent BioAnalyzer 2100 after purification by the AMPure bead. The read sizes were estimated to range between 250–300 bp. One library was prepared per accession. In the last step, equal amounts of quantified libraries were pooled to be sequenced on the Illumina HiSeq 4000 system at the Duke Center for Genomics and Computational Biology with a length of 2 × 150 bp (pair ends).

### 2.5. Data Processing, Transcriptome Assembly and Annotation

Prior to analysis of data, raw reads were processed to filter out low quality nucleotides (<Q30), contaminant sequence, adapters, and short reads (<50 bp). Pre-processing quality was performed by using the Trimmomatic v0.32 program [21] with ILLUMINACLIP:2:30:10, LEADING:5, TRAILING:5, SLIDINGWINDOW:4:30, and MINLEN:50 parameters. Then, all RNA-Seq data were deposited in the NCBI SRA database under the project PRJNA481444.

Only clean reads were used in downstream analysis. De novo transcriptome assemblies were constructed and evaluated with the four following tools: (1) Trinity v2.4.0, using a k-mer size of 25 [22]. (2) Velvet v1.2.10 [23] followed by Oases v0.2.09 [24] with a series of k-mer sizes from 21 to 61 (step size 2). The parameters used with Velvet were “-shortPaired and -read_trkg” and Oases “-min_trans_lgth 200” to set the minimum length for a transcript in the output file to 200 bp. (3) BinPacker v1.0 [25]. (4) Bridger [26]. Similar to Trinity, BinPacker and Bridger programs were used with a k-mer size of 25, because this k-mer size works well for most organisms.

All de novo assemblies were performed with default settings. To evaluate the quality of transcriptome assemblies (assembly completeness), the number of full-length or nearly full-length transcripts was counted. In this case, a sequence homology search was run on all de novo assemblies with BLASTX and the Swiss-Prot/UniProt database [27]. Then, the BLASTX hits were filtered from annotated transcriptomes with an e-value > 1 × 10^−20^. Next, the length coverage percentage for the top matching database entries was examined using analyze_blastPlus_topHit_coverage.pl script from the Trinity package [22]. The detonate score was calculated by DETONATE’s RSEM-EVAL [27] as an additional criterion to evaluate the quality of different assemblies.

TransDecoder and Trinotate software suites were used for functional annotation of each assembly following the method outlined at http://trinotate.github.io/. TransDecoder v2.0.1 (available at http://transdecoder.sourceforge.net/) was used to predict open reading frames (ORFs) of at least 100 amino acids in length. To further maximize sensitivity for capturing ORFs, all ORFs were scanned for homology to known proteins (including a BLASTP search against UniProtKB/Swiss-Prot, and a search of the peptides for protein domains using Pfam). After extraction of ORFs from the assembly, the Trinotate v2.0.2 (available at http://trinotate.github.io) was preferred for assigning functions, as the package uses multiple databases for annotation. The Trinotate annotation pipeline comprises several software packages such as BLASTX, BLASTP, Pfam search, SignalP, and RNAmmer. Initially, transcripts were searched against the Swiss-Prot protein database by running BLASTX, allowing one hit and with the output in tabular format. Then, the expected protein translations obtained using TransDecoder were searched against the Swiss-Prot/UniProt databases. Functional domains were identified using HMMER v.3.1b2 tools and the Pfam domain database [28,29]. Potential signal peptides were predicted using SignalP v4.1 [30]. TMHMM v.2.0c was used for prediction of transmembrane helices in proteins [31] and RNAmmer v.1.2 was applied to predict ribosomal RNA [28]. All results were deposited into the Trinotate-provided SQLite database, and a spreadsheet summary report was generated. WEGO software [32] was used for Gene ontology (GO) functional classification using the Trinotate-integrated UniProtKB GO annotations. Furthermore, a COG screening for phylogenetic classification was performed using predicted proteins and the EggNog database [33], with an e-value threshold of 1 × 10^−5^. TFs were identified and classified using iTAK (http://bioinfo.bti.cornell.edu/cgi-bin/itak/index.cgi), and finally unigenes involved in the monoterpenoid biosynthesis pathways were identified based on the annotation results, and were assessed using BLAST against the NCBI and UniProt databases.

### 2.6. Identification of Orthologous Genes and Comparative Phylogenetic Analysis

As an initial scenario, OrthoFinder was used [34] with the default parameters to identify orthogroups among predicted protein sequences obtained from Transdecoder v2.0.1 for the five assemblies. We also broadened the sampling by including *Salvia splendens* [35] and *Mentha spicata* (Bioproject ID PRJNA359989) as outgroups. Orthogroups with only single-copy genes were retained for further analysis. The nucleotide sequences for each group were collected into a FASTA file and were multiply aligned using MUSCLE v3.8.31 [36]. Output alignment files were further applied to filter out the poorly aligned regions by the trimAl v1.4 [37] with the parameter “-gt 0.8 -st 0.001”. For each alignment sequence, RAxML v8.2.11 [38] was used to reconstruct the maximum likelihood (ML) tree under the GTRGAMMA model with 100 bootstrap replicates. All gene trees were fed into ASTRAL-II [39] to infer the species phylogeny under a multi-species coalescent model. In addition to ML analysis, a Bayesian methodology using BEAST v.2.5.2 [40] was applied as an alternative method to infer the taxa relationship and estimate divergence time based on all single-copy genes. To generate input files for BEAST, the BEAUti interface was used in an HKY model with four γ categories, a strict molecular clock, and a calibrated Yule process of speciation with two calibration points: Mentha-Thymus at 50 MYA (M = 3.88, L = 0.25), and Salvia-Thymus at 30.6 MYA (M = 3.443, L = 0.15). The calibration dates were obtained from TimeTree (http://timetree.org). The chain length was set to 10 million. The trees were then interpreted by the program TreeAnnotator v1.6.1 prior to visualization in the program FigTree v1.4.0 [41].

### 2.7. Positive Selection Analysis

In the positive selection analysis, the corresponding codon-based nucleotide alignments (single-copy genes) for five *Thymus* species were obtained using PAL2NAL [42] with the help of amino acid alignments. Next, a gene tree was constructed by RAxML [38] using a GTR + γ model for each alignment. In order to do tests for positive selection, the branch-site model was implemented in the CodeML program [43] to estimate the dN/dS (non-synonymous vs synonymous) substitution rates (ω value) for each single-copy orthologous sequence individually. The CodeML analysis used an unrooted tree obtained from relative alignment. We executed CodeML four times to stipulate four separate wild species (*T. daenensis*, *T. lancifolius*, *T. persicus*, and *T. pubescens)* as the foreground lineages. Genes were deemed under positive selection if they yielded statistically significant results for the likelihood ratio test (LRT, *p*-value < 0.01) comparison between the null and alternative models. For each positively selected gene (PSG), functional information was deduced based on its ortholog in *Arabidopsis thaliana*. GO enrichment analyses of PSGs were conducted using the web-based agriGO [44] with the TAIR database, the singular enrichment analysis (SEA) method, 0.01 as the significance level, and 5 as the minimum number of mapping entries.

### 2.8. Ks Analysis for Whole Genome Duplication (WGD)

A common approach was used based on pairwise divergence between paralogs and orthologs at synonymous sites (Ks) to check for possible whole genome duplication (WGD) events in the genus *Thymus*. The FASTKs pipeline with default settings [45] was used to construct Ks plots. In the FastKs pipeline, translated transcriptomes obtained using TransDecoder were blasted against themselves and each other (a cutoff of 1 × 10^−40^) to identify putative paralogs and orthologs. Pairs longer than 100 amino acids were aligned using MUSCLE v3.8.31 [36], then the corresponding codon-based nucleotide alignments were obtained using PAL2NAL [42] with the help of amino acid alignments. The Ks value was calculated for each pair using CodeML in PAML v4.8 [43], with the paired sequence settings (yn00) [46] and the F3x4 model [47]. Finally, mclust v5.0.2 [48] was implemented in R to estimate the normal mixture models for Ks values.

## 3. Results and Discussion

### 3.1. Genome Size Variation across the Genus Thymus

Prior to the transcriptome sequencing of the different *Thymus* species, the genome size of the different accessions of the collection was estimated by flow cytometry. The flow cytometry in 18 accessions belonging to 11 species of the genus *Thymus* resulted in a maximum difference of 2.102-fold (Table 1). The smallest haploid genome size was estimated to be 528.25 Mb (1C-value = 0.54 pg) for *T. daenensis*, and the highest value was 1144.54 Mb (1C-value = 1.17 pg) for *Thymus fedtschenkoi*. The haploid genome size for *T. vulgaris* and *Thymus migricus* (West Azerbaijan) were measured to be 802.15 and 596.72 Mbp (1C-value: 0.82 and 0.61 pg), respectively, which is similar to the results from previous reports [49,50]. The haploid genome in two populations of *Thymus kotschyanus*, collected from the same geographical region (Qazvin), was 1066.28 Mbp.

The 1C-value for two populations of *T. lancifolius* varied from 0.56 (Isfahan) to 1.11 (Kordestan), a 1.98-fold difference that shows intraspecific variation in genome size (from 547.81 to 1085.84 Mbp). Two species of *T. fedtschenkoi* and *T*. *pubescens* showed different genome sizes between populations. Within-species genome size variation has been found in samples from geographically separated populations [51], which probably reflects polyploidy in species populations. *T. persicus* is an endemic species that is only found in a restricted region of Iran [52]. The 1C DNA contents and genome size of this species located in West Azerbaijan measured 1.08 pg and 1056.49 Mbp, respectively. In the three accessions of *T. daenensis*, no variation was observed among the 1C-values of populations (0.56, 0.55, and 0.54 pg, referring to Lorestan, Fereydunshahr, and Arak accessions, respectively). The 1C-value given here is consistent with a previous report that describing the DNA content of similar accessions [50], except in the case of the Lorestan population. The changes in genome size may not only be restricted to species divergence but also be associated with various environmental conditions and developmental stages affecting different populations or individual plants [53].

### 3.2. RNA Sequencing and De Novo Transcriptome Assembly

To obtain *Thymus* transcriptome data, paired-end RNA-Seq libraries were constructed for *T. daenensis* (Tdae, from Lorestan), *T. vulgaris* (Tvul), *T. lancifolius* (Tlan, from Kordestan), *T. persicus* (Tper, from West Azerbaijan), and *T. pubescens* (Tpub, from West Azerbaijan). The five libraries described above were represented by 60.98, 64.99, 65.85, 61.81, and 62.20 million reads on Illumina HiSeq 4000, respectively. To facilitate and improve quality and efficiency of assembly, a fraction of reads (28.59% on average, ranging from 22.59% to 32.41%) were removed by filtering adapter sequences and low-quality and short reads (shorter than 50 base pairs). After filtering, a total of 43.38, 43.92, 47.10, 47.62, and 43.38 million high-quality clean reads from the cDNA libraries of Tdae, Tvul, Tlan, Tper, and Tpub were obtained, respectively.

Clean reads obtained from samples were independently assembled de novo using Trinity, Bridger, BinPacker, and Velvet/Oases as described in Section 2, and the resultant assemblies were analyzed to evaluate quality. There are different criteria used to evaluate de novo transcriptome assembly quality, each of which has its advantages and disadvantages. As the first assembly quality metrics, assemblies were evaluated through the number of predicted transcripts and assembly size with different k-mer lengths. Among Trinity and BinPacker with k-mer 25 and Bridger with two k-mers (25 and 27 bp), Trinity produced the highest number of transcripts in all samples (ranging from 85,084 to 123,698) whilst BinPacker and Bridger with k-mer 27 perched on the opposite side. With Velvet/Oases, the total number of transcripts dropped off with an increment in k-mer size, similar to previous studies [54,55], but increased slightly between k-mer 51 and k-mer 57. BinPacker produced a larger assembly size for all samples (ranging from 68.45 to 98.79 Mb) compared to the other programs. Results for the assembly size in Velvet/Oases showed a reversed trend as k-mer size increased. K-mer size plays an important role in increasing or decreasing the number of predicted transcripts and assembly size [55,56]. In theory, large k-mer sizes lead to a decrease in the total number of transcripts, because high k-mer values have a tendency to produce a more contiguous assembly consisting of high-coverage transcripts [56]. This may be due to the capturing of only highly represented reads, or the better assembly of highly expressed transcripts. In contrast, low k-mer values produce a highly fragmented assembly due to sequencing errors and lack of overlap. Short k-mers can lead to the generation of multiple paths in the assembly graph. Therefore, the graph structure can be ambiguous, and the represented isoforms can be challenging to resolve. Because of ambiguous regions, a single path is almost impossible to find in the assembly [22,56]. The results in the present study show that assembly size and the total number of contigs depend on k-mer length, which is highly linked to how the assembler handles the ambiguous paths.

Bridger and BinPacker yielded N50 values ranging from 819 to 1553 bp in Tdae, 732 to 1603 bp in Tvul, 767 to 1634 bp in Tlan, 999 to 1665 bp in Tper, and 818 to 1574 bp in Tpub. BinPacker produced the longest N50 value and had the highest average contig length, while Trinity performed the worst for the N50 length and average contig length in all samples. Bridger had slightly better N50 values in comparison to Trinity, similar to the results from previous research [56]. In Velvet/Oases, the N50 and average contig length slightly increased with a simultaneous increase of k-mer size to k = 29 or k = 31 before deteriorating. However, a longer N50 does not necessarily produce a better assembly, as it may indicate a high level of chimerism [55,57], and previous research indicates that the largest N50 can artificially infer a higher quality of assembly [58,59]. A summary of statistics for all assemblies in all species is shown in Appendix A.

We further evaluated the different assemblies by estimating the number of recovered full-length transcripts when the Swiss-Prot/UniProt protein database was compared with Blastx. In terms of percentage of full-length transcripts, BinPacker generated more full-length transcripts (ranging from 5403 to 6008), with more than 80% coverage for Tdae and Tvul, respectively, while Velvet/Oases was the worst (Appendix A). As in previous studies, our results indicate the Trinity assembler is able to recover more full-length transcripts than Velvet/Oases [21,56,60]. In this case, Bridger assembled more full-length transcripts than Trinity, and was the closest assembler to BinPacker [25,55].

According to the detonate score, all assemblies were very similar except Velvet/Oases (which ranged from −3,716,349,411.54 to −3,787,528,395.51). The assemblies with the highest to lowest RSEM-EVAL scores were as follows: Trinity > Bridger > BinPacker > Velvet/Oases. Assemblies produced by Trinity and Velvet/Oases scored well with the highest (>96%) and lowest (<70%) percentage of RMBT (reads mapped back to transcriptome), respectively. RMBT results showed that BinPacker, Bridger, and Trinity performed similarly (ranging from 92% to 96%).

Based on the evaluation metrics performed, comprehensive comparison showed that BinPacker achieved the best results with a longer assembly size and N50, whereas the total number of contigs, RMBT percentage, and detonate score did not show a significant difference in comparison with Bridger (the assembler that produced the lowest number of contigs) and Trinity (the assembler that produced the highest RMBT percentage and detonate score). Hence, the BinPacker assembly was selected as the best for conducting downstream analyses.

### 3.3. Database Searches and Gene Ontology

We used the BinPacker assemblies for each species as queries and ran BLASTX to search against the UniProt database with no e-value cut-off, but an e-value of 1 × 10^−5^ as the threshold to report the results. A total of 39,968 (percentage of all unigenes in a species: 60.02%), 54,555 (61.33%), 53,922 (56.79%), 41,372 (61.58%), and 44,347 (65.39) unigenes of Tdae, Tvul, Tlan, Tper. and Tpub, respectively, were annotated. Overall, Tpub had a higher percentage of unigenes annotated compared to the other species. The number of predicted proteins ranged from 13,571 to 11,549. Tvul and Tlan had the highest numbers of predicted proteins, possibly because a greater size of sequencing data can provide a better assembly. Furthermore, predicted protein lists with significant sequence similarity were further used to compare the number of shared proteins among species (Figure 1). Five assemblies shared 6158 proteins. The highest number of protein sequence homology (21.5–24%) was shared between Tlan and Tvul (9447 proteins), while Tlan and Tpub shared the lowest number of proteins (8596 proteins), followed closely by Tdae and Tper (8597 proteins).

To categorize the biological functions of transcripts, the contigs having sequence homology with UniProt annotations were given GO (gene ontology) assignments under biological processes, cellular components, and molecular function domains. The distribution of functional categories for all five transcriptomes was visualized using the Web Gene Ontology Annotation Plotting (WEGO) application [32]. The derived graph (Figure 2) indicates that enriched gene ontology term annotations were broadly distributed across the three main domains, and the percentages of sequences mapped to a given sub-ontology were highly similar for all species. Based on the obtained results, most of the GO terms were significantly overrepresented in cellular component domain. Within this category, cell class, followed by cell part and intracellular classes, were predominant. In the biological process domain, the two most common categories were cellular process and metabolic process. As far as molecular function, the most abundant classes were binding and catalytic activity.

Overall, the GO term abundance results revealed that the main GO classifications were metabolism and fundamental biological regulation, and a majority of unigenes involved in terpenoid biosynthesis were found in the categories of metabolic processes. These results are similar to other terpene-producing plant species such as *Salvia miltiorrhiza* [61] and *Melaleuca alternifolia* [62].

According to functional annotations based on the COG databases, unigenes were clustered into 24 COG categories (Figure 3). The largest category was function unknown, followed by posttranslational modification, transcription, and signal transduction mechanisms. Nuclear structures and extracellular structures were the smallest groups. The number of assigned proteins to secondary metabolite biosynthesis ranged from 350 to 409. The presence of genes with unknown functions in these species suggests that the regulation of biosynthesis of secondary metabolites may involve novel factors that require further study.

Based on the KEGG assignment, the majority of predicted protein-coding genes were distributed under the “metabolism” classification (Appendix A). The genes in this category were further classified into 12 subcategories, of which “global and overview maps” were the most prevalent, followed by “carbohydrate metabolism”, while the metabolism of terpenoids and polyketides categories contained a few gene members. This may be because the associated metabolites could be specific in each species. These results agree with transcriptome studies on other medicinal plants (e.g., *Cinnamomum camphora* [63] and *Murraya koenigii* [64]). It was also found that the number of genes in the secondary metabolic pathways related to diterpenoid biosynthesis (ko00904; ranging from 129 to 146 for Tvul and Tper) and terpenoid backbone biosynthesis (ko00900; ranging from 69 to 77 for Tlan, and Tper) were higher than monoterpenoid biosynthesis (ko00902; ranging from 22 to 31 for Tlan and Tvul) and sesquiterpenoid and triterpenoid biosynthesis (ko00909; ranging from 24 to 48 for Tdae and Tpub).

### 3.4. Identification of Transcription Factors

Terpenes are one of the most diverse and the largest known group of secondary plant metabolites [65,66], and are among the most important compounds in thyme [67,68]. Studies in Arabidopsis and transgenic roses have shown the importance of bHLH and MYB TFs in controlling terpenoid production [69,70]. Homologs of bHLH and MYC2 play roles in the regulation of the biosynthesis of sesquiterpenes in *Artemisia annua* [71], *Solanum lycopersicum* [72], and *Arabidopsis thaliana* [69]. TSAR1 and TSAR2 from the bHLH family activate all genes of the precursor mevalonate pathway [73]. MtTSAR and CrBIS TFs activate distinct terpenoid pathways in their species of origin [74]. Members of the WRKY family such as GaWRKY1 in *Gossypium arboreum* [75], SlWRKY73 in *Solanum lycopersicum* [71], OsWRKY76 in rice [76], and GrWRKY7 in *Gentiana rigescens* [77] have been shown to regulate sesquiterpene and terpene biosynthesis. In kiwifruit, the action of NAC TFs regulate monoterpene production [78]. The AP2/ERF family (e.g., CitERF71) has also been shown to regulate sesquiterpenes [79].

To identify TFs, *Thymus* datasets were annotated against all plant TFs. Unigenes encoding TFs were classified into 53 different TF families (Appendix A). In our datasets, bHLH, NAC, MYB-related, ARF, WRKY, ERF, C2H2, C3H, FAR1, and B3 families constituted a large proportion of transcripts encoding transcription factors in all samples, suggesting their potential function in regulating terpenoids biosynthesis (Figure 4). However, evaluation of expression changes and co-expression pattern analysis of these transcription factors are needed to further indicate the possible positive regulators of terpene metabolism and provide information for the possible genetic regulation network of terpenes in *Thymus.*

### 3.5. Thymol Biosynthetic Pathway

Triannotated transcriptome contigs were analyzed and mined for putative genes involved in thymol biosynthesis. Thymol formation is proposed to be through p-cymene and γ-terpinene intermediates [80,81]. Terpenes are derived from two five-carbon precursors, DMAPP (Dimethylallyl diphosphate) and IPP (Isopentenyl diphosphate), generated through the MVA (Mevalonate) and MEP (2-C-methylerythritol-4-phosphate) pathways, respectively. IPP and DMAPP are subsequently converted to GPP (Geranyl pyrophosphate), GGPP (Geranyl geranyl pyrophosphate) and FPP (Farnesyl pyrophosphate). GGDS occurs in both homomeric and heteromeric forms. The homomeric form is composed of a large subunit (LSU) and the heteromer form consists of an LSY and a small subunit (SSU). Monoterpenes are produced from GPP. Genes associated with terminal steps leading to thymol biosynthesis are GGDS, γ-terpinene synthase (GTP) and cytochrome P 450 71 family (Appendix A).

Based on the integrated annotation process, 81, 91, 80, 97, and 94 transcripts were annotated as 11 genes involved in thymol and other monoterpene biosyntheses (Appendix A) in Tdae, Tvul, Tlan, Tper, and Tpub, respectively. For *DXS* (1-deoxy-D-xylulose 6-phosphate (DOXP) synthase), *HDS* (4-hydroxy-3-methyl-but-2-enyl diphosphate (HMB-PP) synthase), *HDR* (4-hydroxy-3-methyl-but-2-enyl diphosphate (HMB-PP) reductase), and *TPS*, there were high contig numbers (9 to 25 transcripts). *DXR* (1-deoxy-D-xylulose 6-phosphate reductoisomerase) and *IDI* (Isopentenyl diphosphate isomerase) had moderate numbers of contigs (3–10), while *MCT* (2-C-methyl-D-erythritol cytidyltransferase), *CMK* (4-diphosphocytidyl-2-C-methyl-D-erythritol kinase), *MDS* (2-C-methyl-D-erythritol 2,4-cyclodiphosphate synthase)**, and *GDS* (Geranyl diphosphate synthase) all had fewer than four contigs. It is reasonable to expect that the differences in contig number reflects differences in transcript abundance. Given this, the MEP pathway would seem to be the major contributor of monoterpene thymol production, given the higher numbers of contigs associated with this pathway.

Due to the high gene expression level of monoterpene synthase genes in leaves, a query-based search showed the presence of complete coding regions for most of these genes (Appendix A). However, IDI, GDS, and some terpene synthases (TPSs) were found as partial sequences in some species. TPSs are highly diversified throughout the plant kingdom and many exhibit tissue-specific expression. Analysis of the several plant genomes indicates TPS genes are a mid-size family, with the gene number ranging from 18 to 152, although not all of them have activity [82]. To date, a number of TPSs including TPS1–TPS9 have been identified and characterized in *T. Vulgaris*, *T. caespititius*, and *T. albicans* [80,83,84,85,86]. Nine to 25 *Thymus* transcripts were identified as putative TPS genes in different species grouped into four TPS classes, including TPS2, TPS3, TPS4, and TPS5. Among the identified TPS genes, the maximum number of unigenes was identified for TPS5, followed by TPS3. TPS5 displays a responsibility for the generation of volatile monoterpenes because of a similar trend with the emission of monoterpenes [87,88]. Through the homology search in the current assembled transcriptomes, the full-length sequences were identified for TPS5 in all species, TPS2 in only Tdae, TPS3 in Tvul, Tlan, and Tper, and finally TPS4 in all species except Tper. Global alignment revealed a >89% similarity between TPS5 and the sequence reported in NCBI (KC461937.1) belonging to *T. vulgaris* with length of 1795 bp. The sequences of unigene candidates encoding terpenoid biosynthesis enzymes are listed in Additional File 1. All four putative TPS sequences displayed high similarity to known plant TPS genes based on a comparison at the amino acid level. The amino acid comparison showed the highest similarity (~93%) between TPS2 and TPS3, followed by TPS4 and TPS5 (~89%). In contrast with this, the sequence of TPS4 displayed low levels of similarity (max. 63%) to TPS2.

TPSs are divided into three subclasses based on their functional roles and product formation, namely, monoterpene synthases (Mono-TPSs), diterpene synthases (Di-TPSs), and sesquiterpene synthases (Sesqui-TPSs). Based on sequence homology, TPSs are classified into seven families, from TPSa to TPSg [66,82]. However, most monoterpene synthase genes, especially in *T. vulgaris*, belong to the Mono-TPSb subfamily [85,89]. Here, we used the TERZYME program [90] to classify TPSs. As anticipated, all putative TPSs identified in the *Thymus* transcriptomes were assigned to the Mono-TPSb subfamily, supporting the existing view that *Thymus* TPSs belong to a distinct clade. All the discovered TPSs were investigated for the presence of terpene synthaseconserved motifs. The deduced amino acids of all TPSs contained the DDXXD motif, which is a highly conserved motif in TPSs and plays a role in the complexation of the diphosphate group after the ionization of substrates [82,91]. To obtain further information, phylogenetic analysis was performed to distinguish the evolutionary relationships of terpene synthases. Phylogenetic analysis placed TPS2 and TPS3 in a clade (Figure 5), while TPS5 was determined in a different clade. In the tree, it was observed that all TPSs identified in this study clustered into the TPS-b subfamily, which consisted mainly of Mono-TPSs. Phylogenetic analysis shows that TPSb is closely related to TPSg, which is congruent with previous phylogenetic studies of the TPS gene family [81,91]

TPSs along with CYPs organize the core components of terpene biosynthetic pathways [92]. It has been reported that the terminal gene involved in thymol biosynthesis is a regio-, stereo-, -specific Cytochrome P 450, specifically a member of the CYP71D family [93]. One previous study suggests that TPS genes are predominantly found in combination with CYP71 clan genes in both eudicots and monocots, and that among the TPS/CYP pairings there is a significant correlation between the TPSb and CYP71D subfamilies [92]. In the *Thymus* transciptome data, at least 45 putative CYP450s transcripts were identified belonging to five families including CYP71D, CYP72A, CYP736A, CYP749A, and CYP82d, with the majority of CYP716B2 family members. Thus, from this data we can conclude that the CYP genes may be involved in thymol and carvacrol biosynthesis.

### 3.6. Orthogroups Identified, Phylogenetic Analysis and Finding Genes under Positive Selection

Our understanding of the phylogenetic relationships of Lamiaceae has improved in recent years, with new phylogenetic studies placing Lamiaceae within a large clade called “core Lamiales” [94] that comprises more than 7000 species assigned to 236 genera [2]. However, the relationships among most genera and species have remained unresolved. In the present study, a large-scale phylogenetic reconstruction of *Thymus* species, which has been investigated so far mainly using small-scale molecular data [95], was carried out using 592 single-copy genes as follows.

An orthogroup is the set of genes that have evolved from a single gene of the last common ancestor of all the species via speciation [96]. To identify the orthogroups for the five *Thymus* species, with *S. splendens and M. spicata* as outgroups, protein sequences obtained from assembled and annotated transcriptome data were used with OrthoFinder [34]. The total number of orthogroups with all species present was 592. These were used for a subsequent phylogenetic analysis to define evolutionary relationships between these species.

Relationships found in our ML tree for the most part agree with the Bayesian tree (Figure 6). These phylogenetic trees strongly support a close genetic relationship between *T. daenensis* and *T. vulgaris* as sisters, as well as between this clade and a clade composed of *T. pubescens* and *T. lancifolius*.

Interestingly, the close relationship between *T. daenensis* and *T. vulgaris* is strongly supported by morphological and oil content similarity. *T. daenensis* is characterized by a high percentage of thymol, and *T. vulgaris* displays a mixture of thymol and g-terpinene as major constituents [97].

Divergence time estimation resulting from a Bayesian analysis suggests that *Thymus* originated during the late Pleistocene (2.24 MYA), then diverged and dispersed. This estimate falls into the range of ages estimated by Drew et al. (2012) [98]. Phylogenetic analysis of 121 accessions of the tribe Mentheae based on cpDNA and nrDNA suggest that *Thymus* originated during the Mid-Pliocene (3.06 MYA, between 2.40 and 3.72 MYA) [98].

In a ML framework, divergence dates generally resulted in an increase in the estimated ages of younger nodes. The ML tree indicated that *Thymus* originated during the Tortonian (10.2 MYA). This difference is due to the methodology used for time calibration [99], molecular substitution [100], and clock model choice [101].

Orthogroups shared between *Thymus* species ranged from 18,412 between Tdae and Tper, and 21,071 between Tlan and Tvul. There were also 727 single-copy orthogroups in our species comparison that were retained for positive selection analysis. Using the branch-site model, 55, 49, 29, and 54 genes were identified as being under positive selection in Tdae, Tlan, Tper, and Tpub, respectively. Positively selected genes were further used to identify functional categories according to their assigned gene ontologies. The distribution of classification of PSGs showed that in all species (Appendix A), biological process (GO:0008150) had the most hits. The second most abundant GO category was related to metabolic processes (GO: 0008152).

The unigenes subjected to positive selection were further annotated in the Swiss-Prot database (Appendix A). Our search for gene matches from this study with other research revealed remarkable similarities. A protein involved in the pathway protein ubiquitination, namely F-box protein, was found to be common among all species. Indeed, it has been indicted that some F-box proteins are under selection and show a strong tendency of positive selection [102,103]. F-box genes show various roles in plants, including diverse abiotic and biotic responses and flowering and growth regulation [104]. Another group of positively selected genes, PPR (pentatricopeptide repeat), which has been found in all species, performs essential roles in plant embryogenesis and chloroplast RNA metabolism [105]. In many cases, PPR proteins are coded by restorer of fertility (Rf) genes that suppress cytoplasmic male sterility (CMS). The interaction of CMS genes and Rf genes leads to gynodioecy, which appears to be a stable sexual system in *Thymus* [106]. These Rf-PPR genes show a high dN/dS ratio in different species [107,108]. Our results also revealed that some genes belonging to the bHLH, MYB, and MADS-box TF families might also be positively selected and have roles in the diversification of these species.

### 3.7. Whole Genome Duplication Analysis

To identify WGD, first, the synonymous mutation rate (Ks) was estimated for each paralogous pair, then gene pairs showing abnormally high (>3) or low (<0.02) Ks values were discarded to make plots. As shown in Figure 7, the Ks values of paralogous genes showed an L-shaped distribution in all five *Thymus* transcriptomes, and no significant second peak was observed in the Ks-distributions. Overall, our result revealed that the small-scale gene duplications were dominant over long-term period (Ks < 3) and most of the paralogous genes were probably newly duplicated, which agrees well with the concept that most duplications are young. This small-scale duplication may also be associated with the genes related to secondary metabolites, and perhaps can explain the widespread monoterpene diversification [109,110]

## 4. Conclusions

In this study, we presented annotated transcriptomes for five different *Thymus* species using mRNA sequencing (RNA-seq), providing valuable resources for studying molecular mechanisms of terpenoid biosynthesis, the development of molecular markers, and evaluating phylogenetic relationships among species. This also led to the identification of putative genes involved in thymol biosynthesis. Most notably, three unigenes (germacrene D synthase, γ-terpinene synthase, and a member of the cytochrome P450 71 family) were identified as probable terminal steps of thymol biosynthesis. GO, COG, and KEGG enrichment analysis revealed that numerous contigs encode enzymes involved in secondary metabolic pathways. The functional characterization identified key TPS genes that were clustered into the TPS-b subfamily. The phylogenetic tree based on single-copy orthogroups showed a close genetic relationship between *T. daenensis* and *T. vulgaris*. Between 50 and 60 genes showed evidence of positive selection in the *Thymus* lineages. The present analysis based on L-shaped Ks distribution indicated that WGD is probably absent in *Thymus* species, or that WGD has probably not contributed substantially to duplicated genes in these species, and that polyploidy does not appear to be responsible for the large genome size in *T. lancifolius*, *T. persicus*, or *T. pubescens*. The study also provides worthy resources for a bioengineering study of terpenoids in *Thymus* species with pharmacologically increased metabolite content.

## Figures and Tables

**Figure 1 genes-10-00620-f001:**
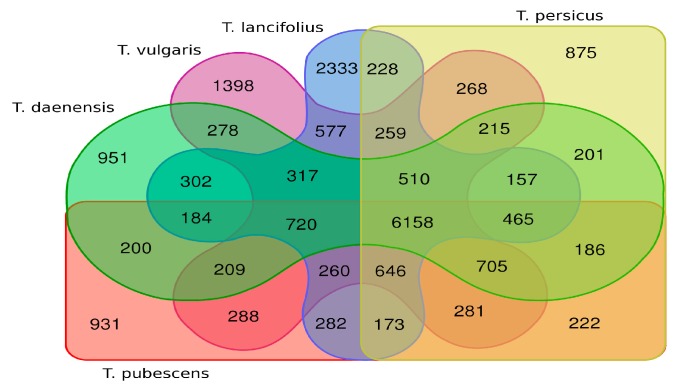
Venn diagram showing the number of overlapping proteins between the five *Thymus* transcriptome assemblies.

**Figure 2 genes-10-00620-f002:**
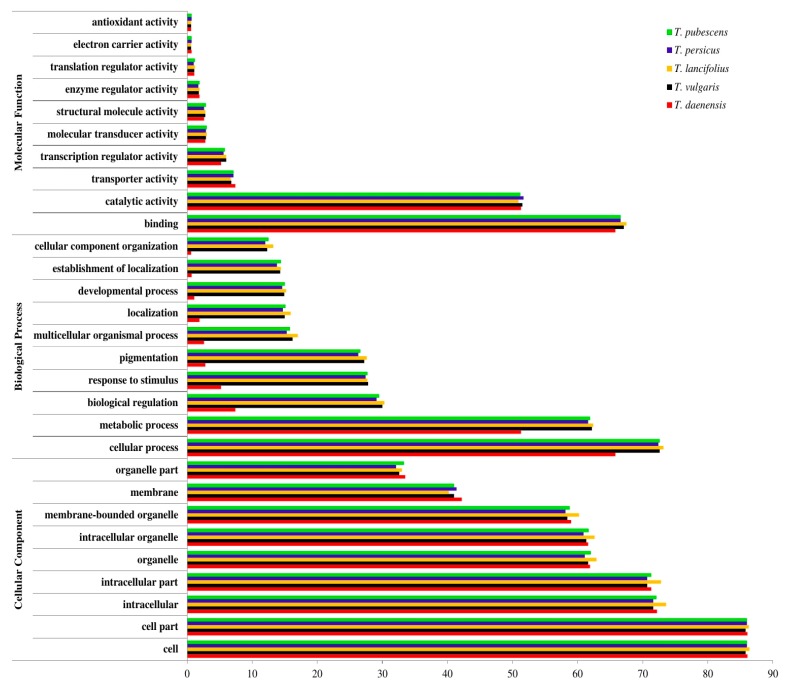
Gene ontology (GO) classification map derived from *Thymus* transcriptomes. The results are summarized in three GO categories: biological process, molecular function, and cellular component. The x-axis indicates the percentage of sequences in each category.

**Figure 3 genes-10-00620-f003:**
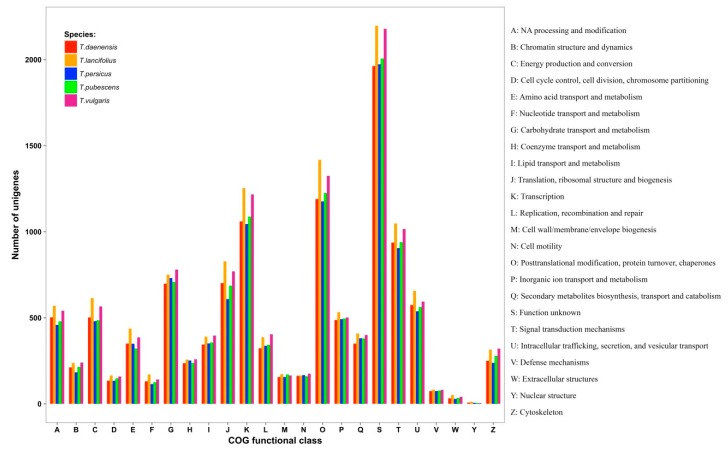
COG classification map. The abscissa represents 24 COG group names, while the vertical axis represents the number of unigenes annotated into each group.

**Figure 4 genes-10-00620-f004:**
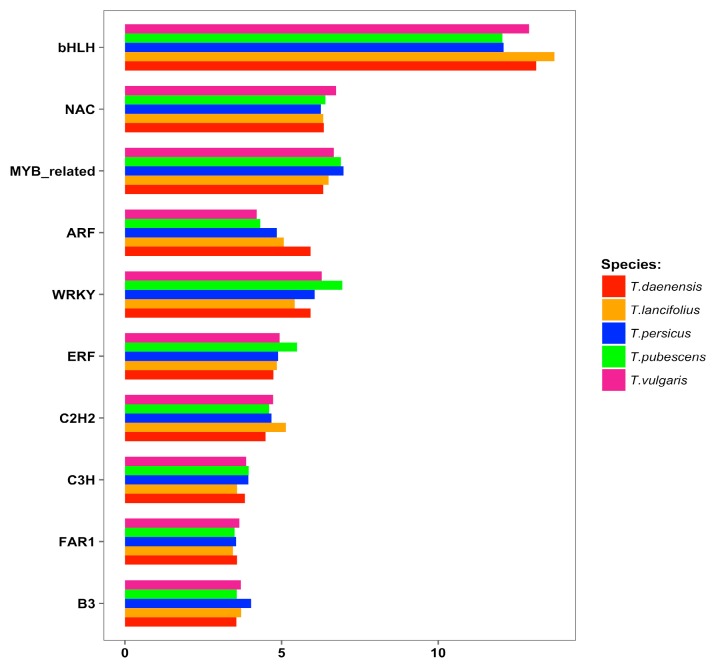
Distribution of transcripts in different transcription factor (TF) families showing over-representation in bHLH.

**Figure 5 genes-10-00620-f005:**
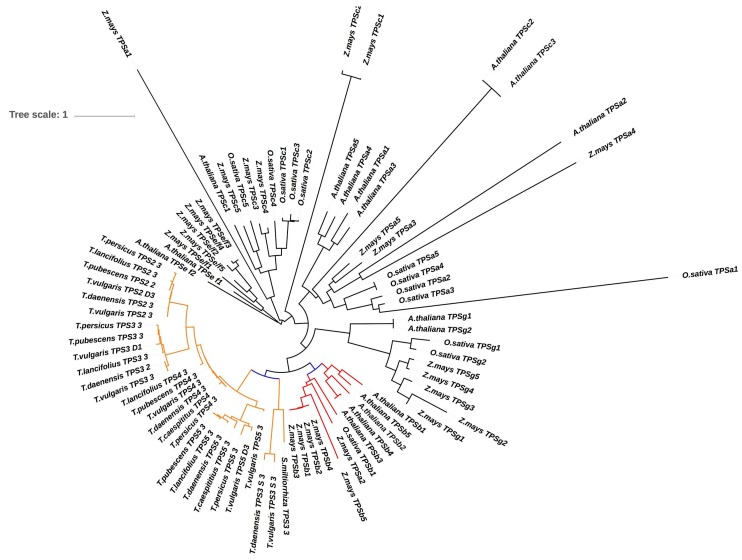
Maximum likelihood (ML) phylogeny of putative TPS genes in *Thymus* species compared to other species of TPS subfamilies.

**Figure 6 genes-10-00620-f006:**
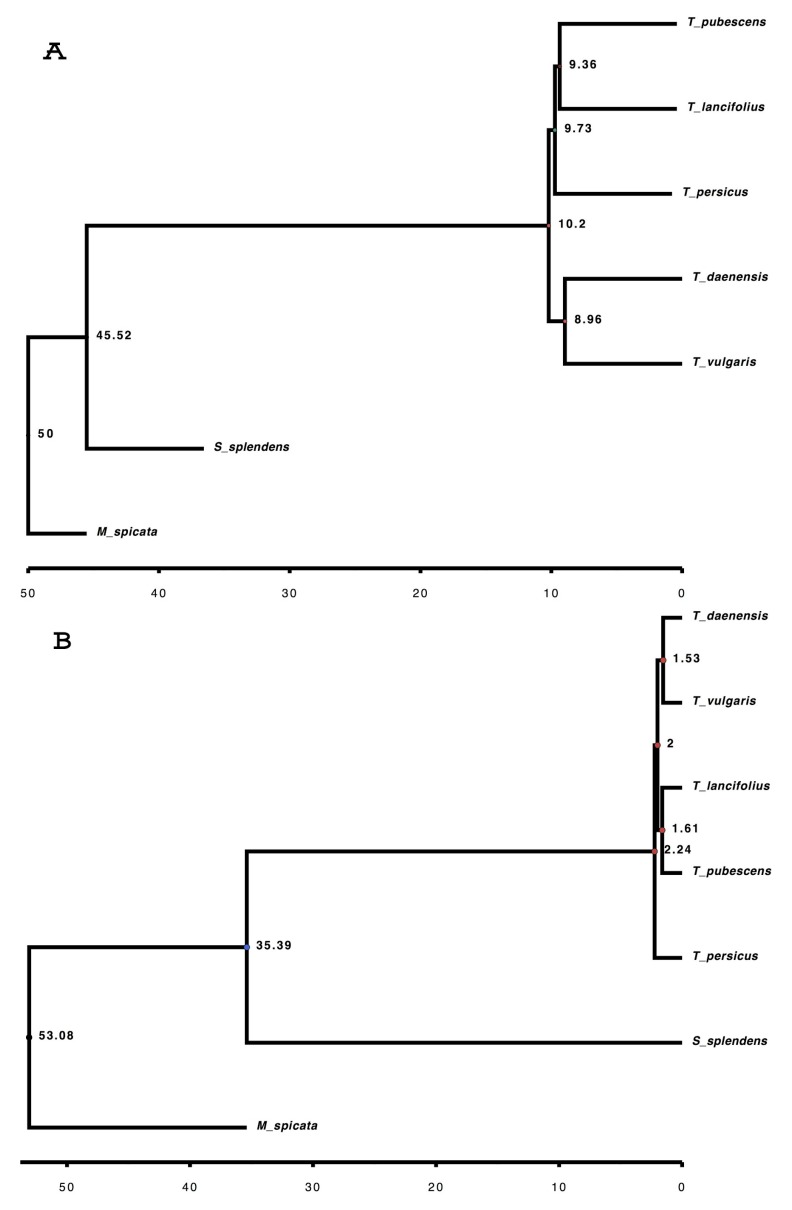
Phylogenetic relationships among five *Thymus* species inferred from 592 single-copy orthogroups. (**A**) Maximum likelihood tree. (**B**) Bayesian tree.

**Figure 7 genes-10-00620-f007:**
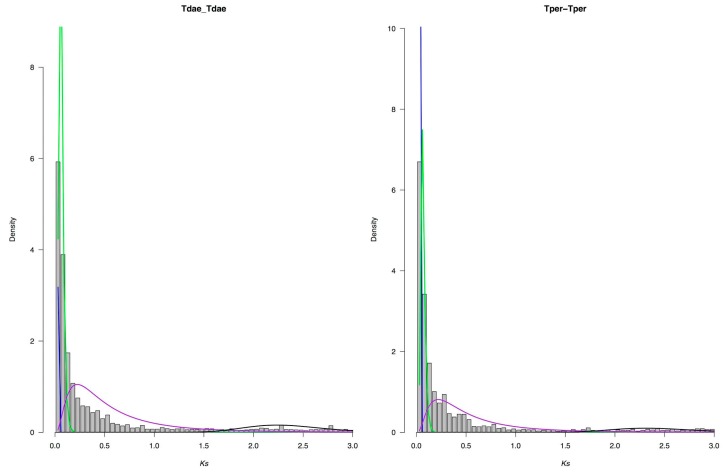
Ks distribution plots for *T. daenensis* and *T. persicus.* The components are (1) background gene duplications (green), (2) putative whole-genome duplications (purple), (3) alleles or sequencing errors resulting in high identity transcripts (blue), and (4) potential whole-genome duplications with low support (black).

**Table 1 genes-10-00620-t001:** Mean (three replicate measurements per plant sample) 2C-value and genome size of *Thymus* accessions.

Species	Location (Province)	1C-Value ± SD (pg)	Haploid Size (Mbp)	Estimated Polyploidy
*Thymus pubescens*	East Azerbaijan	0.63 ± 0.02	616.14	2n = 2x
*T. pubescens*	West Azerbaijan	1.11 ± 0.01	1085.84	2n = 4x
*Thymus migricus*	West Azerbaijan	0.61 ± 0.01	596.72	2n = 2x
*Thymus daenensis*	Lorestan	0.56 ± 0.01	547.81	2n = 2x
*T. daenensis*	Fereydunshahr	0.55 ± 0.27	538.03	2n = 2x
*T. daenensis*	Arak	0.54 ± 0.28	528.25	2n = 2x
*Thymus persicus*	West Azerbaijan	1.08 ± 0.01	1056.49	2n = 2x
*Thymus kotschyanus*	Qazvin	1.09 ± 0.05	1066.28	2n = 2x
*T. kotschyanus*	Qazvin	1.09 ± 0.04	1066.28	2n = 2x
*Thymus lancifolius*	Isfahan	0.56 ± 0.01	547.81	2n = 2x
*T. lancifolius*	Kordestan	1.11 ± 0.04	1085.84	2n = 4x
*Thymus vulgaris*	Cultivated	0.82 ± 0.02	802.15	2n = 2n
*Thymus fedtschenkoi*	Semnan	0.61 ± 0.01	596.72	2n = 2x
*T. fedtschenkoi*	Yazd	1.17 ± 0.02	1144.54	2n = 4x
*Thymus transcaspicus*	Khorasan	1.09 ± 0.02	1066.28	2n = 2x
*T. transcaspicus*	Yazd	1.07 ± 0.02	1046.71	2n = 2x
*Thymus fallax*	Yazd	1.14 ± 0.02	1115.19	2n = 2x
*Thymus carmanicus*	Kerman	1.08 ± 0.02	1056.49	2n = 2x

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
