# Peer review of "Transcriptome Landscape Variation in the Genus Thymus"

_genes, 2019, doi:10.3390/genes10080620_

Round 1

Reviewer 1 Report

The manuscript presents a transcriptomic study for 5 species belonging to an important and rich genus of aromatic plants. However, in my opinion it is too descriptive and doesn’t corresponds to the final statement on the Abstract,” which was to provides a global overview of genes related to terpenoid pathway in Thymus and can help to understand relationship among the Thymus species”. Also, as a general comment, the authors should be more rigorous in their statements and revise carefully the literature they cite to support the Discussion.

Furthermore, the manuscript here and there needs more rigor, such as in the Abstract and Introduction, because sometimes there are confusions on Thymus as a genus and the species T. vulgaris. On L 14, when the authors state: “Thymus has been known as an economically important herb for its medicinal and aromatic properties.” Actually, Thymus is not a herb, it is a genus with several species, several of which have interesting qualities. Further down on the text, on L 37, the authors state “Diverse studies have investigated the chemical composition and biological properties of essential oils in different species of Thymus [4]”, but reference 4 is for T. vulgaris, the authors should choose another reference, more adequate to what they are saying.

It is not clear how many species the authors studied. On 2.1. Plant Material, this should be better explained, because they refer to table 1, which presents 11 species for DNA content study, but going to the end of Introduction and Results section, NGS only addresses 5 species. This should be explained on the methodology. What is the rationale for choosing these 5 species?

Some incorrections are present: on L 370-372, the authors state “They contain characteristic motifs seen in monoterpene synthases. Specifically, the N-terminal domain has conserved double arginine motif RR(X)8W, and an aspartate rich domain DDXXD motif is present in C terminal domain [78].” Reference 78 is “Mendes, R.S.; Von Atzingen, M.; de Morais, Z.M.; Gonc ̧ales, A.P;, Serrano, S.;M;, Asega, A.F;, Romero, E.C.; Vasconcellos, S.A; Nascimento, A.L. The novel leptospiral surface adhesin Lsa20 binds laminin and human plasminogen and is probably expressed during infection. Infect Immun 2011, 79, 4657–4667.”; this reference is not adequate. There is information published for monoterpene synthases in thymol-rich Thymus species, T. vulgaris and other Thymus species, and the authors should revise the literature with attention and then improve here the Discussion. Actually, the manuscript doesn’t cite important and recent literature.

Finally, on the conclusion section, L441-443 the authors write: In this study, we presented annotated transcriptomes for five different Thymus species using mRNA sequencing (RNA-seq), providing valuable resources for studying molecular mechanisms of terpenoid biosynthesis; my question is what’s the true contribution of this study for helping the understanding of terpenoid metabolism?? Were there isogenes for terpene synthase genes? How many putative TPS did they find for the different species? What is the similarity among them? How conserved are the sequences?

On L444, they state “Large scale transcriptome sequencing data provided a comprehensive overview of gene expression pattern related to terpenoid biosynthesis and led to the identification of putative genes involved in thymol biosynthesis.”, where are these genes, I mean where is this data? This information is not provided on the supplementary material, neither on the data presented on the paper. I cannot see the identification of putative genes neither the comparison with other described genes available for Thymus that are published. All the information about the transcription factors is also unexploited, not really developed.

As a final comment, one main objective of this research was, as far as I could understand, to mine putative genes involved in thymol and carvacrol biosynthesis, but I question if this goal has been achieved!!

Other corrections proposed:

The values with the Mean presented on Table 1 should have a s.d. (Standard deviation)

Author Response

Reviewer #1:

>> We appreciate that the reviewer’s comments. The followings are our point-by-point responses:

Comment 1: On L 14, when the authors state: “Thymus has been known as an economically important herb for its medicinal and aromatic properties.” Actually, Thymus is not a herb, it is a genus with several species, several of which have interesting qualities. Further down on the text, on L 37, the authors state “Diverse studies have investigated the chemical composition and biological properties of essential oils in different species of Thymus [4]”, but reference 4 is for T. vulgaris, the authors should choose another reference, more adequate to what they are saying.

>> Response: Thank you for pointing out the wrong use of Thymus word. We have modified the text accordingly (line 36 to 39) adding other reference in which other Thymus species are described from a biochemical point of view.

Comment 2: It is not clear how many species the authors studied. On 2.1. Plant Material, this should be better explained, because they refer to table 1, which presents 11 species for DNA content study, but going to the end of Introduction and Results section, NGS only addresses 5 species. This should be explained on the methodology. What is the rationale for choosing these 5 species? Explained

>> Response: You have raised an important point here. We apologize for not making it clear. We added more information and explained why 5 species were selected for NGS (Section 2.3).

Comment 3: Some incorrections are present: on L 370-372, the authors state “They contain characteristic motifs seen in monoterpene synthases. Specifically, the Nterminal domain has conserved double arginine motif RR(X)8W, and an aspartate rich domain DDXXD motif is present in C terminal domain [78].” Reference 78 is “Mendes, R.S.; Von Atzingen, M.; de Morais, Z.M.; Gonc ales, A.P;, Serrano, S.;M;, Asega, A.F;, Romero, E.C.; Vasconcellos, S.A; Nascimento, A.L. The novel leptospiral surface adhesin Lsa20 binds laminin and human plasminogen and is probably expressed during infection. Infect Immun 2011, 79, 4657–4667.”; this reference is not adequate. There is information published for monoterpene synthases in thymol rich Thymus species, T. vulgaris and other Thymus species, and the authors should revise the literature with attention and then improve here the Discussion. Actually, the manuscript doesn’t cite important and recent literature.

>> Response: Thank you for point out the wrong reference and the inadequate discussion of the results. We removed not adequate references and cited recent researches. Then we have changed some sentences and improve discussion. We hope that this version meets the quality standards.

Comment 4: Finally, on the conclusion section, L441-443 the authors write: In this study, we presented annotated transcriptomes for five different Thymus species using mRNA sequencing (RNAseq), providing valuable resources for studying molecular mechanisms of terpenoid biosynthesis; my question is what’s the true contribution of this study for helping the understanding of terpenoid metabolism?? Were there isogenes for terpene synthase genes? How many putative TPS did they find for the different species?

On L444, they state “Large scale transcriptome sequencing data provided a comprehensive overview of gene expression pattern related to terpenoid biosynthesis and led to the identification of putative genes involved in thymol biosynthesis.”, where are these genes, I mean where is this data? This information is not provided on the supplementary material, neither on the data presented on the paper. I cannot see the identification of putative genes neither the comparison with other described genes available for Thymus that are published. All the information about the transcription factors is also unexploited, not really developed.

>> Response: We agree with the reviewer. We have extended our analysis to identify terpene synthase genes, specially TPSs. More information about the number of contigs, genes length and phylogenetic analysis was provided to the part of  “3.5 Thymol biosynthetic pathway” .  Additionally we have changed our conclusions. Now it read as: “In this study, we present annotated transcriptomes for five different Thymus species using mRNA sequencing (RNA-seq), providing a valuable resource for the study of the terpenoid biosynthesis in the Thymus genus, the development of molecular markers, and evaluating phylogenetic relationships among species”. We hope that this information would improve the revised version of the manuscript

Comment 5:  The values with the Mean presented on Table 1 should have a s.d. (Standard deviation)

>>Response: As suggested by the reviewer, standard error was added as fourth columns of table 1.

Reviewer 2 Report

Comments to the Author:

In this manuscript the authors have conducted de-novo transcriptome assembly and analysis of five Thymus species. For obtaining best assembly they compared results of four assemblers and then compared the sequences of the assembled transcripts with various databases to obtain annotations. They also showed phylogenetic relation between the species and identified genes under selection. This study allows the authors to study terpenoid biosynthesis and generate a resource of markers for future in-depth studies.  My comments are described below.

## Introduction ##

1: Line 34-35.

‘Most of the species have originated in the Mediterranean region, and since has been distributed worldwide’. Is this information included in the citation [1,2] ?

2: Line 47.

It feels to me that discovery is stressed  rather than sequencing of genome , transcriptome, etc, in the words ‘through the discovery of genomes, …’.

3: Line 49:

‘fragments’ should be ’reads’

4: Line 45-51

In this paragraph you describe NGS and its benefits in general and in agriculture.  But it lacks necessary citations of NGS reviews and instances of where NGS gave a breakthrough in agriculture or evolutionary science.

### Results and Discussion ### 

5: Line 226.

‘quality assembly metrics’ should be ‘assembly quality metrics’

6: Line 240:

’The results in the present study that assembly size’ should be ‘The results in the present study shows that assembly size’

7: Line 241.

Could you please explain what do you mean by ‘small ambiguities’ here.

8: Line 241.

‘low coverage paths’.  You talk here and above (line 237) about ‘coverage’.  But you have not shown any statistic about the coverage or depth of the sequencing. So do you have any data about the coverage or depth of sequencing to correlate with the statement you make?

8: Line 251.

In the table S1, why  have you not included information about Velvet/Oasis ?

9: Line 251 : Table S1.

Column name ‘L50’ should be ’N50’

10: Line 273-274:

‘during the report results’ should be  ‘to report the results’

11: line 279:

‘Interestingly’ should be ‘Furthermore’

12: Line 323.

When you say ‘These results agree with the transcriptome studies in other medicinal plants’, do you mean all the annotation results of this study?

And which results do you mean. ? Is it that most the genes cluster in global and overview maps in KEGG or function unknown category in COG ?

13: Line 343.

‘FAR’ should be ‘ARF’ transcription factor

14: Line 357.

Did you generate this figure S2?

15: Line 391.

‘Interetsingly’  should be ‘Interestingly’

16: Line 391.

You talk about close relationship between T. deanensis and T. vulgarise in phylogeny. So do the proteins they code, overlap more than other species. ?

17: Line 407.

Here you mention ‘727 single copy orthogroups’.  But in abstract section, line23, you mention ‘592 single copy orthologous’.  Are these genes/orthogroups different ?

18: Line 430 Figure 6

Figure 6 legend does not show what different coloured lines and curves represent.

The legend shows that it is frequency plot, but in the figure the y-axis says density.

19:  Why the sequenced data is not submitted in online repository?

20: Line 487.

This line and line 496 repeated.

Author Response

Reviewer# 2

>> We are grateful to the reviewer for his/her insightful comments on my paper. Thank you!

Comment 1: 1: Line 3435. ‘Most of the species have originated in the Mediterranean region, and since has been distributed worldwide’. Is this information included in the citation [1,2] ?

>> Response :We agree with this comment. A more appropriate reference (3) was added. 3- 3.            Morales, R. The history, botany and taxonomy of the genus Thymus. In: Stahl-Biskup., Saez, F. (Eds), Thyme: The Genus Thymus. Taylor & Francis, London. pp: 1-43. 2002.

Comment 2:  2: Line 47. It feels to me that discovery is stressed rather than sequencing of genome , transcriptome, etc, in the words ‘through the discovery of genomes, …’.

>> Response : We agreed with the reviewer. We revised the sentence to emphasize point.

Comment 3:  3: Line 49: ‘fragments’ should be ’reads’

>>Response: We polite disagree with the reviewer. The term “read” can be applied only to the DNA (or RNA) fragments that have been sequenced. So, the correct term when a library is prepared is “fragment” or more specifically for a cDNA library “cDNA fragment” (https://en.wikipedia.org/wiki/Read_(biology))

Comment 4:  4: Line 4551 In this paragraph you describe NGS and its benefits in general and in agriculture. But it lacks necessary citations of NGS reviews and instances of where NGS gave a breakthrough in agriculture or evolutionary science

>>Response: We agree with the reviewer. Therefore, we added two more adequate references.

Comment 5: Line 226. ‘quality assembly metrics’ should be ‘assembly quality metrics’

>>Response ; Thank you for pointing this out. This term has been revised.

Comment 6: Line 240: ’The results in the present study that assembly size’ should be ‘The results in the present study shows that assembly size’

>> Response ; Thank you for pointing this out. We have changed accordingly.

Comment 7: Line 241. Could you please explain what do you mean by ‘small ambiguities’ here.

>> Response ; The text was changed. We hope that now it is clear.

Comment 8: Line 241. ‘low coverage paths’. You talk here and above (line 237) about ‘coverage’. But you have not shown any statistic about the coverage or depth of the sequencing. So do you have any data about the coverage or depth of sequencing to correlate with the statement you make?

>> Response ; The text was changed to make it clear.

Comment 9: Line 251. In the table S1, why have you not included information about Velvet/Oasis ?

>> Response ; As suggested by the reviewer, we provided information about Velvet/Oasis in supplementary file

Comment 10: Line 251 : Table S1. Column name ‘L50’ should be ’N50’

>> Response ; Thank you for pointing this out. Revised!

Comment 11: Line 273274: ‘during the report results’ should be ‘to report the results’

>> Response ; Thank you for pointing this out. Revised!

Comment 12:  line 279: ‘Interestingly’ should be ‘Furthermore’

>> Response ; Thank you for pointing this out. Revised!

Comment 14: Line 343. ‘FAR’ should be ‘ARF’ transcription factor

>> Response : Thank you for pointing this out. We have changed accordingly.

Comment 15: Line 391. You talk about close relationship between T. deanensis and T. vulgaris in phylogeny. So do the proteins they code, overlap more than other species. ?

>> Response : The close relationship is based in a phylogenetic study of the five different species that we present in this manuscript, no in the “overlap” between the proteins (that it could be an ambiguous term).

Comment 16: Line 407.Here you mention ‘727 single copy orthogroups’. But in abstract section, line23, you mention ‘592 single copy orthologous’. Are these  genes/orthogroups different ?

>> Response : Thank you for pointing this out. Mentha and Salvia species were removed for positive selection analysis, in this case the number of SCGs was increased .

Comment 17: Line 430 Figure 6 Figure 6 legend does not show what different coloured lines and curves represent. The legend shows that it is frequency plot, but in the figure the yaxis says density.

>> Response : You have raised an important point here. We provided an explanation to this figure to make it clear.

Comment 18: Why the sequenced data is not submitted in online repository?

>> Response : All RNA-Seq data were deposited in the NCBI SRA database under the project PRJNA481444

Round 2

Reviewer 1 Report

The manuscript has improved considerably with the revision. It enriches the knowledge in the field of secondary metabolism genes in aromatic plants, which is an interesting subject. In this new version, the authors performed a more robust discussion and brought new and updated references to the manuscript. However, some papers were not included and should be cited because they are related to the area. On L415-416, when the authors state “To date, a number of TPSs including TPS1 trough TPS9 have been identified and characterized in T. vulgaris and T. caespititius (80, 82 84); the authors should include a recent publication by Filipe et al (Journal of Plant Physiology 218 (2017) 35–44) on T. albicans. Also, terpene synthase genes has recently been revised in Trindade et al (2018) Industrial Crops & Products 124 (2018) 530–547, and this reference should also be included.

Following these corrections, I think the paper can be accepted.

Minor comments:

Ref. 84 is incomplete.

On Table 1, No SD is presented, although the authors replied to this comment.

Furthermore, on that table, under the column Estimated Polyploidy, correct ploidy units, they should be n, not X.

L114- Correct were

Author Response

The manuscript has improved considerably with the revision. It enriches the knowledge in the field of secondary metabolism genes in aromatic plants, which is an interesting subject. In this new version, the authors performed a more robust discussion and brought new and updated references to the manuscript.

>> We would like to thank the reviewer for her/his general evaluation

However, some papers were not included and should be cited because they are related to the area. On L415-416, when the authors state “To date, a number of TPSs including TPS1 trough TPS9 have been identified and characterized in T. vulgaris and T. caespititius (80, 82 84); the authors should include a recent publication by Filipe et al (Journal of Plant Physiology 218 (2017) 35–44) on T. albicans. Also, terpene synthase genes has recently been revised in Trindade et al (2018) Industrial Crops & Products 124 (2018) 530–547, and this reference should also be included.

>> Thank you for the suggestions. Both references have been included. Trindade et al. 2018 is the reference 81 (cited at line 387) and Filipe et al. 2017 is the reference 86 (cited at line 406).

Following these corrections, I think the paper can be accepted.

Minor comments:

Ref. 84 is incomplete.

>> We have completed the ref. 84. 

On Table 1, No SD is presented, although the authors replied to this comment.

>> We apologize. The SD has been added to this new version.

Furthermore, on that table, under the column Estimated Polyploidy, correct ploidy units, they should be n, not X.

 >> We politely disagree with the reviewer in this point. n, indicates the number of gametes, so a haploid cell is noted as n and a somatic cell as 2n. x, indicates the number of chromosome’s sets, so a diploid species is noted as 2n = 2x = number of chromosomes (e.g. Solanum lycopersicum will be 2n = 2x = 24). A tetraploid species will be noted as 2n = 4x (e.g. Nicotiana tabacum will be 2n = 4x = 48).

L114- Correct were

>> Corrected.